# Regenerative Potential of Hydroxyapatite-Based Ceramic Biomaterial on Mandibular Cortical Bone: An *In Vivo* Study

**DOI:** 10.3390/biomedicines11030877

**Published:** 2023-03-13

**Authors:** Katarína Vdoviaková, Andrej Jenca, Andrej Jenca, Ján Danko, Lenka Kresáková, Veronika Simaiová, Peter Reichel, Pavol Rusnák, Jozef Pribula, Marko Vrzgula, Sarah J. Askin, Maria Giretová, Jaroslav Briancin, Lubomír Medvecký

**Affiliations:** 1Department of Morphological Disciplines, University of Veterinary Medicine and Pharmacy in Kosice, Komenskeho 73, 041 81 Kosice, Slovakia; 2Solmea s.r.o., Bacikova 7, 040 01 Kosice, Slovakia; 3Educational, Scientific and Research Institute AGEL, Palisady, 743/56, 811 06 Bratislava, Slovakia; 4Svine Clinic, University of Veterinary Medicine and Pharmacy in Kosice, Komenskeho 73, 041 81 Kosice, Slovakia; 5Hospital AGEL Kosice-Saca, Lucna 57, 040 15 Kosice-Saca, Slovakia; 6Department of Anatomy, Pavol Jozef Safarik University in Kosice, Srobarova 2, 041 80 Kosice, Slovakia; 7Ashmore Veterinary Centre, 59 High Street, Geen Street Green, Orpington BR6 6BQ, UK; 8Division of Functional and Hybrid Systems, Institute of Materials Research of SAS, Watsonova 47, 040 01 Kosice, Slovakia; 9Institute of Geotechnics of SAS, Watsonova 45, 040 01 Kosice, Slovakia

**Keywords:** biomaterial, bone defect, lower jaw, ceramic implants, mandible, pig, regeneration

## Abstract

Reconstruction of bone defects and maintaining the continuity of the mandible is still a challenge in the maxillofacial surgery. Nowadays, the biomedical research within bone defect treatment is focussed on the therapy of using innovative biomaterials with specific characteristics consisting of the body’s own substances. Hydroxyapatite ceramic scaffolds have fully acceptable phase compositions, microstructures and compressive strengths for their use in regenerative medicine. The innovative hydroxyapatite ceramics used by us were prepared using the tape-casting method, which allows variation in the shape of samples after packing hydroxyapatite paste to 3D-printed plastic form. The purpose of our qualitative study was to evaluate the regenerative potential of the innovative ceramic biomaterial prepared using this method in the therapy of the cortical bone of the lower jaw in four mature pigs. The mandible bone defects were evaluated after different periods of time (after 3, 4, 5 and 6 months) and compared with the control sample (healthy cortical bone from the opposite side of the mandible). The results of the morphological, clinical and radiological investigation and hardness examination confirmed the positive regenerative potential of ceramic implants after treatment of the mandible bone defects in the porcine mandible model.

## 1. Introduction

In the 21st century, the very fast lifestyle and current technological progress is leading to breakthroughs in the development of biomedical research and medicine generally, resulting in improvement in the diagnoses of serious illnesses and financially more effective treatments. Bone defects of the mandible can develop due to various causes, such as ablative tumour therapy, diverse traumatic injuries (vehicle accidents, firearms, burns, scalds, electrical flashes), congenital malformations (orofacial clefts), bone cysts, inflammatory infectious conditions or osteonecrosis (avascular, post-radiation, after antibiotic therapy) [1]. Damage to the maxillofacial area (except bone defects) is closely related to numerous soft and hard tissue injuries (eyes, nasal airways, paranasal sinuses, tongue), neurologic and vascular involvement and avulsion of the temporomandibular joint and supporting structures [2,3].

Reconstruction of the bone defects and maintaining the continuity of the mandible is still a challenge in maxillofacial surgery, especially with respect to the preservation of the basic masticatory functions and swallowing, speech articulation and phonetic functions, supporting the teeth and deglutition. Some discontinuities of the mandible also result in significant aesthetic, cosmetic and functional changes to a part of the face due to the loss of the anatomical mandibular shape. It is necessary to restore the mandible bony continuity and facial contour to maintain tongue mobility and restore the innervation of the affected facial area. After tumour ablation or complex resection of the mandibular segmental parts, the damaged structures must be reconstructed and restored for rehabilitation [4]. According to the localisation and extent of the mandible bone defects, they can generally be divided into defects of the rostral and lateral part of the mandible and of the ramus mandible [5]. Classification of the mandibular defects according Adelusi [6], describes segmental or marginal bone defects of the mandible. This division depends on resection of part of the mandible leaving the unaffected part intact. Marginal defects mean resection of the mandibular part without a continuity break, whereas in the case of segmental defects the bone integrity is not ensured.

The current potential for regenerative medicine is supported in two ways. The first way consists of mesenchymal stem cells therapy and the second is about tissue engineering with the use of natural or synthetic materials. Stem cell therapy is suitable for healing of smaller areas with bone defects. Innovative tissue engineering is a good strategy for the healing and reconstruction of the bigger segmental bone defects [7,8]. At present, modern technologies such as 3D printing, bioengineering, mesenchymal stem cells or bone osteosynthesis using mini plates are considered revolutionary ways for treating bone defects in the maxillofacial area [9,10,11].

The options for treatment of mandible defects include the use of autologous microvascular free flaps, bridging plates and transport distraction osteogenesis and bone grafts. The current experimental medicine strives to develop other alternative treatment methods based on special innovative biomaterials that may succeed in ideal reconstruction of mandible bone defects [12]. Nowadays, biomedical research on the treatment of bone defects of the face focusses on therapy using innovative biomaterials with specific characteristics consisting of the body’s own substances.

Calcium phosphate or HAP scaffolds generally have a large utilization in medical practice, therefore form another option as a suitable material for the treatment of mandible bone resection in dentistry surgery [13]. Biomaterials based on calcium phosphate have been clinically established in maxillofacial surgery as bone and defect fillers. They have a very specific osteoinductive, osteoconductive and proliferative potential, supporting differentiation of individual cell types [14,15]. Utilization of the special porous calcium phosphate scaffolds results from their favourable mechanical and physicochemical properties, surface texture and microstructure characteristics, such as particle size and pore size distribution. The HAP ceramic scaffolds have fully acceptable phase composition, microstructure and compressive strength, which are advantageous in reconstructive or regenerative medicine. Acellular scaffolds based on hydroxyapatite must serve more biological functions, e.g., migration and proliferation of new specific cells, biodegradation, biocompatibility, bioactivity, osteoconductivity and osteoinduction, formation of new bone tissue and osteointegration. A very important property of calcium phosphate ceramics is the wide pore size distribution, which is favourable from the point of view of new bone tissue formation [16]. Proliferation of the osteoblasts can be strongly affected by the high concentrations of calcium and phosphate ions after amorphous calcium phosphate dissolution. In in vitro studies, osteoblast bone cells were attached and adhered to the surface of the HAP scaffold where they proliferated and secreted extracellular matrix without formation of the any toxic products or effects. The initial bone cell spreading and cytoskeletal organization were related to the crystallinity of the HAP material. The detachment strength of the osteoblast cells was related to surface roughness, fibronectin preadsorption and integrin subunit sensitivity. An in vitro study showed that the short- and longer-term responses of bone marrow cells were improved with the roughness of the HAP material [17].

In the preclinical research focused on regeneration of the mandibular bone defects by using innovative biomaterials, it is desired to employ a large animal as an experimental model [18]. Swine may serve as a very good animal model for the testing of bone implants because the bone mineral density, morphology, remodelling rate and healing are similar to that in humans [19]. Immature and mature swine skeletons have individual growing phases with very well-developed Haversian systems [20]. Porcine models were used in the studies of the osteogenic potential of the biomaterials in the maxillofacial area of the head and in dental surgery involving teeth implants [21,22].

The purpose of our qualitative study was to evaluate the regenerative potential of innovative hydroxyapatite-based ceramic biomaterial prepared using the tape-casting method in the remodelling of the cortical bone of the lower jaw in four mature pigs.

## 2. Materials and Methods

### 2.1. Material Fabrication

#### 2.1.1. The Synthesis of HAP Powder

Hydroxyapatite powder was synthesized by the precipitation of 0.5 M Ca(NO_3_)_2_·4H_2_O (SIGMA-Aldrich, Merck KGaA, St. Louis, MO, USA, analytical grade) solution and 0.5 M (NH_4_)_2_HPO_4_ (Sigma-Aldrich, Merck KGaA, St. Louis, MO, USA, analytical grade) solution with a molar ratio of Ca/P equal to 1.66. The aqueous solution of Ca^2+^ ions was slowly dropped into phosphate aqueous solution for 1.5 h using a peristaltic pump at a pH close to 10.5 by adding NH_3_(aq) (1:1) at 25 °C. The rotation speed of magnetic stirrer was 450 rpm. Precipitates were aged for 72 h at room temperature with a 24 h exchange of solution above HAP layer. HAP precipitates were washed with distilled water, then ethanol and then filtered. Hydroxyapatite powders were dried at 110 °C for 4 h, then crushed and sieved (Mesh 250).

#### 2.1.2. The Preparation of the HAP Ceramics

Hydroxyapatite ceramics was prepared by the tape-casting method, where synthesized HAP was mixed with (2 wt% carboxymethyl cellulose + 1 wt% polyacrylic acid) hydrogel + glycerol for 5 min. in planetary ball mill (Fritsch, agate balls and vessel). The ratios of hydrogel to glycerol and HAP/hydrogel were 6:1 and 1:1, respectively. The paste was moulded to polylactic acid forms, printed using a 3D printer (3D printer da Vinci 3 in 1, XYZ printing Inc., Thailand), dried at 37 °C for 12 h, removed from the form and dried at 100/1 h and 120 °C/1 h. Final green samples were sintered at 1250 °C for 2 h; the heating rate was 5 °C/min. The properties of the HAP ceramic biomaterial used in our study are presented in Table 1.

### 2.2. In Vivo Creation of the Mandibular Defect

#### 2.2.1. Animal Model

The material prepared by the above-described method was implanted into the mandibular defects of four female adult pigs (breeding farm PD Agro Michalovce, Slovak Republic) of the Large White breed, which had an average weight of 258.9 kg ± 3.57 g (mean ± standard deviation) (Table 2). The experimental right side of the mandible was compared with the healthy cortical bone from the left side (at the same level of teeth) that served as a control. The animals were housed at the Swine Clinic at the University of Veterinary Medicine and Pharmacy in Kosice (Kosice, Slovakia) and the health status of the animals was monitored daily throughout the experiment. All pigs had *ad libitum* access to standard food and water. The experiment was approved by the State Veterinary and Food Administration of the Slovak Republic (Study No. 4650/17-221).

#### 2.2.2. Premedication and General Anaesthesia

Prior to surgery, the experimental animals were pre-medicated by intramuscular administration of the mixture of atropine (0.02 mg/kg body weight) (Atropine sulphate, 1 mg/mL solution for injection, Fatro s.p.a, Bologna, Italy), azaperone (5 mg/kg body weight) (Stresnil 40 mg/mL, Janssen Pharmaceutica, Beerse, Belgium) and butorphanol (0.02 mg/kg body weight) (Butomidor 10 mg/mL, Richter Pharma, Wels, Austria). The general anaesthesia was established with thiopental (15 mg/kg body weight) (Thiopental VUAB 1.0 g, VUAB Pharma a.s., Czech Republic) administered intravenously.

#### 2.2.3. Clinical Procedures

Using an extra-oral approach to the mandible, unilateral defects were created in the cortical bone. The surgical incision was made on the right side of the mandible at the level of the first and second premolar teeth. The marginal bone defects on the right side of the exposed mandible body were created using a reciprocating bone saw cooled with 0.9% sterile saline. The created defects were 1.6 cm long (anterior–posterior), 0.8 cm wide (buccal–lingual) and 0.4 cm thick (inferior border–height of contour). A sterile hydroxyapatite ceramic plate with length 1.5 cm, width 0.7 cm and thickness 0.3 cm was inserted to fill the marginal defect on the right side of the mandible (Figure 1). The presence of blood from the surrounding bone tissue was very important for stimulation of the defect healing. The native bone on the left side of the mandible body was used as a control. After the surgical procedure, resorbable sutures were used to close all muscles layers and skin, and the closed wound was covered by liquid aluminium bandage. The conventional radiographs of the lateral view of the porcine mandible were obtained post-surgery to confirm the correct inserting of the ceramic plate.

#### 2.2.4. Post-Operative Care

The experimental animals were fed liquid feed during the first week after surgery; then, they were provided a standard diet (the same as the one fed to them during the pre-operative period) and had free access to water during convalescence. After the surgery, the general health of animals was observed closely every day, including determination of body temperature, checking of the surgical wound, faecal consistency, signs of infection or any other complications. The broad-spectrum antibiotic oxytetracycline dihydrate (1 mL/10 kg body weight) (Alamycin LA a.u.v., Norbrook, Newry, UK) was administered intramuscularly every second day for 7 days of post-operative care to control potential infections. For pain and inflammation prevention, an anti-inflammatory drug flunixin meglumine (2.2 mg/kg body weight) (Flunixin a.u.v., Norbrook, Newry, UK) was injected intramuscularly once a day for 7 days.

#### 2.2.5. Animal Euthanasia and General Observations

Following the convalescence period, the animals were euthanized singly after different periods of time (after 3, 4, 5 and 6 months) in accordance with the animal protection laws. After sedation of the animals with azaperone (2 mg/kg body weight) (Stresnil 40 mg/mL Janssen Pharmaceutica, Beerse, Belgium) intramuscularly, euthanasia was performed intravenously with thiopental (90 mg/kg body weight) (Thiopental VUAB 1.0 g, VUAB Pharma a.s., Czech Republic).

The skin and muscles were removed from the previous right side mandibular defect site and thus the newly created bone was exposed. The appearance of the new bone surface was evaluated macroscopically. The bone surface was examined, digitally photographed and harvested for documentation. The healed mandible bone defect site was analysed macroscopically, histologically, immunohistochemically and by radiographic imaging. The left side of the mandible body was used as a control.

### 2.3. Evaluation of the Biomaterial and Newly Formed Bone Tissue

#### 2.3.1. Assessment of Bone Hardness

Samples from experimental animals and the HAP ceramic (standard) embedded in a resin were polished with SiC paper (Struers, 320, 800, 1200, 2500 and 4000) and used for hardness measurement by an indentation test (Microhardness TUKON 1102, Wilson hardness, Shanghai, China). Vicker microhardness was measured at three different distances (0.5, 1.5 and 3.5 mm; marked as microH 500, 1500 and 3500, respectively) from the bone surface with the implanted HAP ceramic. The microhardness of the HAP ceramic was measured in the middle of the thickness. The average value (10 measurements for each distance) for the selected area was expressed as the Vickers degree of hardness (HV). A 100 g load was applied to the surface of samples using a diamond indent for 15 s. Statistical evaluation of the results (n = 10) was performed by ANOVA analysis. A value of *p* = 0.05 was considered significant.

#### 2.3.2. XRD Phase Analysis, Compressive Strength and Microstructure of Samples

The green samples had plate shapes (2.0 × 1.0 × 0.3 cm). The apparent density of sintered ceramic samples was determined from measurement of the dimensions and mass of the samples. The theoretical density of HAP was 3.15 g·cm^−3^. The phase composition of samples was analysed by X-ray diffraction analysis (XRD, Philips X PertPro). The microstructure of the scaffolds was observed using scanning electron microscopy (SEM, (JEOL FE SEM JSM-7000F). The compressive strength of the ceramic samples was measured by a LR5K Plus (Lloyd Instruments, Ltd., Bognor Regis, UK) at the loading rate of 1 mm/min^−1^.

#### 2.3.3. *In Vitro* Cell Cultivation, Cytotoxicity and Viability Testing

The ceramic samples were sterilized in a thermostat at 160 °C/1 h. The substrates in the form of pellets (~6 mm in diameter and 0.5 mm in height) were soaked in phosphate buffer solution (PBS) at 37 °C for 5 min and placed into the wells of a 96-well suspension plate (Santa Cruz Biotechnology, Santa Cruz, CA, USA). MC3T3E1 Subclone 4 (ATCC CRL-2593) (Manassas, VA, USA) preosteoblastic murine cells at subconfluency were harvested from culture flasks by enzymatic digestion (trypsin–EDTA solution, Sigma). The cells were suspended in a culture medium and the cell suspension was adjusted to a density of 5 × 10^4^ cells/mL. A total of 200 μL of the complete culture α-modification minimum essential Eagle medium (αMEM) (10% foetal bovine serum, 1% antibiotics-antimycotics (ATB-ATM) and osteogenic supplements: β-glycerophosphate 10 mM, ascorbic acid 50 μgmL^−1^ and dexamethasone 50 nM (obtained from Sigma)) containing 1.0 × 10^4^ MC3T3E1 cells were carefully seeded onto each disc.

Wells with cells cultured in culture medium without sample were considered as negative controls. The culture plates were cultivated in an incubator at 37 °C, 95% humidity and 5% CO_2_. The culture medium was refreshed thrice a week. After both 48 h and 10 days, the density, distribution and morphology of the MC3T3E1 cells on the surfaces of tested samples was evaluated by live/dead staining of cells on the samples using an inverted optical fluorescence microscope (Leica DM IL LED, blue filter). Live/dead staining was based on fluorescein diacetate (FDA)/propidium iodide (PI). The cytotoxicity was evaluated according to ISO 10993-5:2009 (ISO 10993-5. ISO 10993-5 Biological Evaluation of Medical Devices—Part 5: Tests for in vitro cytotoxicity. Geneva, Switzerland: International Organization for Standardization 2003) uing a commercially purchased MTS test (Cell titer 96 aqueous one solution cell proliferation assay, Promega, USA) after 48 h and 10 days of cultivation. The intensity of formazan absorption in the culture medium was measured using an UV/VIS spectrophotometer (Shimadzu 1800, Japan) at a wavelength of 490 nm, and mean absorbance (n = 5, single measurement from each of five wells) ± standard deviation was calculated using the Statmost statistical program. The relative viability of the cells was calculated as the ratio of measured absorbances of formazan in wells with sample to the absorbance of formazan in the negative controls.

#### 2.3.4. Macroscopic Assessment

The defect site at the mandible bone and the adjacent soft tissue were evaluated after euthanasia. The macroscopic evaluation of each animal head included the symmetry of the facial part of the head at the site of the created bone defect, colour and signs of inflammation of the skin and surrounding muscles. The defect site was examined for the presence of secretion and formation of the structure of muscle layers. Macroscopic examination of the bone tissue at the site of the healed defect involved monitoring of colour, surface, filling of the defect and the integration of the newly formed bone tissue with the surrounding bone. In addition, the mandible was examined as a whole for the presence of potential bone growth or other pathological changes in the bone.

#### 2.3.5. Histological and Immunohistochemical Assessment

The collected specimens of the mandible (1.0 cm long, 1.0 cm wide and 1.0 cm thick bone blocks) were fixed in neutral formalin for one week, decalcified in chelaton for one month, dehydrated through a series of 70–100% ethanol solutions and embedded in paraffin. The specimens were then serially sectioned at 7 μm thickness in the sagittal plane, mounted on slides and prepared for histological (haematoxylin and eosin staining) and immunohistochemical analysis. The physiological bone sections from the left side of the mandible were used as reference control samples.

An immunohistochemical reaction was performed to demonstrate the presence of collagen I using a primary rabbit polyclonal anti-collagen I antibody (Abcam, ab233080) and a secondary DB DET SYS kit, the DB detection kit—rabbit/mouse dual system (Biotech). DAB (3,3′-diaminobenzidine) (DAKO) was used to visualize the reaction. Finally, the cell nuclei were stained with acidic Mayer’s haematoxylin.

#### 2.3.6. Radiological Assessment

Following a post-surgical period, conventional radiographic examination and computer tomography (CT) of the mandible body after euthanasia were carried out to assess the regeneration of the cortical bone tissue.

The radiographs in the lateral view of the porcine head were obtained using an X-ray machine (Philips Digital Diagnost, Delft, The Netherlands).

CT scans of the head were obtained with a computed tomograph (Philips Brilliance 40-slice CT, the Netherlands) in the axial and sagittal plane. They were used to evaluate neoformation of the bone tissue on the mandible.

## 3. Results

### 3.1. Bone Hardness

Figure 2 shows the values of Vicker microhardness (microH) at different distances from the surface of the bone with the implanted HAP ceramic. The microhardness results were not statistically significant (microH 500 vs. 1500: *p* > 0.483; microH 500 vs. 3500: *p* > 0.924) and were close to the bone microH (middle area) at a tangential distance of 1 cm from the site of implantation (53.1 ± 6.1 HV). It should be noted that the slightly lower microH 1500 could result from higher bone porosity in this area. It is clear that the more than four times higher microH of the ceramic HAP implants (standard, 229 ± 27.5 HV) compared with the microH of the regenerated bone tissue indicates complete resorption of the ceramics and the formation of new bone tissue. In addition, the newly formed bone had almost the same microhardness as the native bone, so a structural similarity between the new and the original bone tissue can be assumed.

### 3.2. Microstructure, Compressive Strength and XRD Phase Analysis of Ceramics

The dense microstructure of the ceramic sample is shown in Figure 3. In the microstructure, a low fraction of 0.5–2.5 µm spherical pores is visible in the image, which are located mainly at the boundaries between three adjacent grains and formed by coalescence during sintering. In addition, grain boundaries are relatively difficult to distinguish on the fracture surfaces with a transgranular fracture mode, but from close observation we believe that the HAP grain size did not exceed 5 µm. However, the sintering process was stopped before the final phase of densification of the ceramics, where individual micropores are gradually eliminated from the microstructure. The relative density of the HAP ceramic samples reached 84 ± 3% of the theoretical HAP density.

The XRD phase analysis (Figure 4) identified biphasic calcium phosphate ceramics with hydroxyapatite as the main phase ((JCPDS 72-1243) and α-tricalcium phosphate (αTCP) as the secondary phase (JCPDS 29-0359). This composition can correspond to a small chemical non-stoichiometry of HAP powder during synthesis (e.g., as the result of the substitution of carbonates for hydroxyl or phosphate groups), with the formation of calcium-deficient HAP. The average compressive strength of the samples was 71 ± 5 MPa, which is much higher than cancellous bone (up to 15 MPa) but lower than compact bone (up to 200 MPa). Nevertheless, its value is sufficient for utilization of the ceramic as a bone defect filling material.

### 3.3. In Vitro Cytotoxicity of Samples and Live/Dead Staining of Cells

The comparison of relative viabilities (Figure 5) of osteoblasts seeded on ceramic samples clearly demonstrates their non-cytotoxic character because the values of the relative viabilities of cells exceed 100% of negative control.

The results from live/dead fluorescence staining are in accordance with the results obtained from the cell proliferation test. The cells growing on ceramics were well spread, adhered and uniformly distributed on surfaces after 2 days of cultivation; after 9 days, a dense multilayer of osteoblasts covered the entire ceramic surface due to growth of the cell population. Note that no dead cells were found after culture for 2 and 9 days (Figure 6). The cells have a prolonged morphology and long filopodia that mutually interconnected adjacent cells. These facts demonstrate the non-cytotoxic character of the ceramic surfaces, the appropriate texture of them and the physicochemical properties of the samples. In addition, αTCP is more soluble than hydroxyapatite, which can significantly stimulate the osteogenic activity of osteoblasts in contact with the biphasic ceramic system.

### 3.4. Animal Pig Model

During the post-operative period until euthanasia of animals, the general condition of all animals was satisfactory. In one animal was observed a swelling of the soft tissue in the area of the surgery wound in the first two days. In the post-operative period, no animals showed wound dehiscence, macroscopic signs of inflammation, infections nor any other complications at the defect site, and no morphological abnormalities were detected. The implantation of the hydroxyapatite ceramic plate was tolerated well by all pigs used in our study. No loss of body weight was observed during the healing period.

### 3.5. Macroscopic Assessment

The macroscopic examination confirmed that the mandibular wound had healed well in all examined animals after the respective observation periods (3th, 4th, 5th and 6th month). The facial part of the head in all experimental animals was symmetrical. The macroscopic assessment showed that the new bone tissue had an ivory white colour and integrated well with the adjacent native bone. The mandible bone margins at the defect site were easily identified by observation and palpation (Figure 7A–D). Examination of the cross section of the new bone indicated that the healing process resulted in uniform filling of the defect with tissue (Figure 8A,B), which was characterized by a fairly smooth and homogeneous surface. No residues of the ceramic plate were observed at the defect site.

### 3.6. Histological and Immunohistochemically Assessment

Histological analysis of the regenerated mandible bone at the marginal defect site was conducted to look for signs of the presence of calcified bone, fibrous tissue stage and cellular activity. This microscopic evaluation showed differences in the intramembranous bone regeneration in the stage of the bone remodelling according to the length of the post-operative period. No macroscopic defects were found after the respective healing times (3th, 4th, 5th, 6th month). The new tissue in all treated marginal mandible defects extended deep into the defect space. The microscopic analysis allowed us to observe a significant regeneration of compact (cortical) bone into mature, spongy bone.

This observation led to additional analysis of the bone remodelling details in individual time periods during the investigation of the regeneration of the mandible bone. We observed regenerating bone tissue with cells (osteoblasts, osteocytes) and the mineralized matrix. The bone surface was relatively smooth and no ceramic plate residues were observed. The ceramic plate was completely degraded, and the defect was completely filled with new, regenerated bone tissue. The new bone tissue began to form from the margin as a lamellar bone and progressed to the centre of the defect, where the callus was replaced by a compact bone. According to the stage of bone development, the immature bone (woven bone tissue) filled only a portion of the space at the defect site. Resorption by osteoclasts and new bone formation by osteoblasts is typical of bone remodelling. In the zone of the immature bone was observed a bone matrix with a great deal of osteoblasts, disarranged collagen fibres and cement lines. The bone matrix with cement lines was characteristic of the immature, developing bone tissue (Figure 9A–C), whereas osteons with typical arrangement of the osteocytes and the Haversian system in the middle predominate in the mature bone (Figure 9D,Da).

The histological analysis showed that in all time periods of the bone remodelling phase a continuous conversion of the immature to mature bone was observed. In the sample of the bone tissue examined 6 months after the surgery, the presence of bone lamellae, osteons and the Haversian system in the cortical bone was observed. The development of the cortical bone progressed from the perimeter of the bone defect to the trabeculae of the spongy bone. This bone consisted of small cavities filled with bone marrow and fatty tissue (Figure 9D). Histological analysis allowed us to observe the formation of blood vessels and Haversian canals during the healing process.

The immunohistochemical analysis of the samples from the cortical bone defects in the mandible treated using ceramic plates showed the presence of collagen I in the bone matrix of the compact bone, in the spongy bone and around the bone marrow cavities (Figure 10A–D). Figure 10Da shows the detail of the expressed collagen I within and around the osteons.

### 3.7. Radiological Assessment

Computer tomography (CT) and conventional radiographic methods also appeared to be useful tools for the evaluation of the treated mandible bone defects.

The axial and sagittal plane of the CT scan showed reconstruction of the newly formed cortical bone in the area of the originally implanted ceramic plate. The original mandible defects were filled with newly formed mineralized bone tissue, with an average density of 1599 HU that was comparable with the surrounding bone. The values of bone density in each experimental animal are shown in Figure 11. Evaluation of the defects in the sagittal (Figure 12A,B) and axial (Figure 12C,D) plane revealed complete integration of the newly formed bone tissue, with no visible changes in the structure or integrity of the examined bones. The mandible, as a whole showed, no visible macroscopic osteolytic manifestations. The results of the CT examination confirmed the conclusions reached by the morphological and radiographic methods.

Examination by radiological methods revealed the formation of cortical bone in the healed animals. Observation of the area of the treated bone defect on the mandible body using a lateral view confirmed complete integration of the newly formed bone with the surrounding tissue (Figure 13A–D).

## 4. Discussion

Reconstruction surgery of mandible bone defects presents a significant clinical problem in maxillofacial medicine, mainly due to the aesthetic and functional requirements [23]. In this preclinical study, we qualitatively examined the regenerative potential of ceramic hydroxyapatite plates for mandible bone defect repair in pigs. The properties of the ceramic implants and the healing process of the mandible bone defects were evaluated after different periods of time (3, 4, 5 and 6 months).

The pig was used as the most suitable animal model for the treatment of bone defects because pigs demonstrate similarities with human bone formation, function, healing characteristics, mineral density and bone mineral concentration [24,25]. The currently used surgical methods of mandible reconstruction require multiple surgeries to replace the bone at the defect site. The pig lower jaw may help to improve experimental medicine techniques for coordination of the teeth and mandible reconstructions, serving as a model for providing basic improvements to these methods for clinical use in human medicine in the future [26]. The pig lower jaw is also very important for studying of the mechanisms and processes that regulate the new bone formation during desmogenous ossification [27].

In vitro studies are not able to simulate the complex healing process under difficult conditions (e.g., biomechanical properties). Many previous in vitro experiments have shown the positive reaction of bone cells to calcium-phosphate-based materials. In particular, the high concentrations of calcium and phosphate ions can positively affect the proliferation of bone cells [28]. The mature bone cells, osteoblasts, adhered to the ceramic material and were able to proliferate and produce extracellular matrix without any toxic products [29]. The initial cell attachment was affected by alkaline phosphatase activity that favoured surfaces with smooth texture [17]. In the later phases of bone cell adhesion, the surface of the biomaterial played a very important role. This statement was confirmed by another study that described a higher number of osteoblasts attached to rough osteo-ceramic surfaces [30].

The bone healing process in pigs is the same as that in humans [31,32]. Many previous studies indicate that the biochemical properties of the newly formed bone were similar to those of the surrounding bone tissue, as the bone is a dynamic structure undergoing constant remodelling. Each restoration of the fracture or bone defect treatment depends on the material used for the treatment and the treated area of the body; the type of ossification is also very important [33]. Morphological and clinical evaluations of the investigated bones showed no signs of inflammation or infection.

The porous calcium phosphate ceramics have many interesting properties such as biocompatibility, bioresorbability and the ability to support osteoconduction, osteoinduction and osseointegration and have a big potential for the replacement of new bone tissue after maxillofacial surgery. Ceramic implants are considered biodegradable, non-cytotoxic and capable of temporary functioning as a support for the cells or other additives to facilitate the appropriate repair processes. The utilization and application of ceramic materials in reconstructive medicine of the maxillofacial area are determined by their mechanical and physicochemical properties and microstructure characteristics (particle size and pore size distribution) [16,34]. Ceramics based on calcium sulphate have a special structure that permits growth of vasculature and formation of new bone tissue. It is worth mentioning that borophene and its allotropics are used in biomedical applications and could also be used as a prosthetic coating on many different biomaterials [35]. The XRD phase analysis identified a biphasic calcium phosphate ceramic with hydroxyapatite as the main phase ((JCPDS 72-1243) and α-tricalcium phosphate (αTCP) as the secondary phase (JCPDS 29-0359). This composition can correspond to a small chemical non-stoichiometry of HAP powder during synthesis with formation of calcium-deficient HAP. The average compressive strength of the samples was 71 ± 5 MPa, which is much higher than that of cancellous bone (up to 15 MPa) but lower than that of compact bone (up to 200 MPa [36]. According to our results, good adherence and spreading of osteoblastic cells on the ceramic surfaces was observed, and no dead cells were found after culture for 2 and 9 days. The growth of the cell population was clearly visible from the comparison of the 2- and 9-day cultured samples, where a dense layer of osteoblasts covered the entire ceramic surface. These facts demonstrate the non-cytotoxic character of the ceramic surfaces as well as the appropriate texture and the physicochemical properties of the samples. Note that αTCP is more soluble than hydroxyapatite, which can significantly stimulate the osteogenic activity of osteoblasts in contact with biphasic ceramic system. This is also confirmed by the authors’ conclusions in [37]. The outcomes of our study confirmed the support of osteogenic activity and regenerative potential of the HAP implants in the mandible cortical bone with cooperation of cells obtained from the bone marrow through the blood. The osteoblastic cell proliferation was described on the ceramic material surface. A similar activity of stem cells was pointed out in research to evaluate the behaviour of DPSCs grown on silicon nanoporous and mesoporous matrices [38].

There has been a very interesting idea of using a conditioned medium and its products in hard tissue regenerative medicine for a long time. The conditioned medium or secretome of the cells consists of many different but helpfully body substances (signalling and intracellular proteins, enzymes, growth factors, cytokines and hormones). Extracellular vesicles are structures that are released by budding from the plasma membrane into biological fluids or into the culture. Exosomes are deriving from multivesicular bodies and can be considered to be innovative and smart theranostic tools [39].

Resorption of the ceramic plate took about 1–3 months. The density and hardness of the mandibular bones were investigated as an indicator of the healing of the bone defect and the embedding of the ceramic plate in the bone tissue. Our results indicated comprehensive bone development and ceramic plate integration in all experimental animals, such as that declared with other methods used. Biodegradation and resorption of this type of material is faster than the new bone formation [33,40]. The selection of an appropriate therapy for mandible bone defect is challenging due to the presence of the sensory structures, arteries and nerves, lymphatic vessels, features of the face bones and muscles. Maxillofacial surgery must take into the account the restoration of the aesthetics of the face, the function of each structure and the control of bacterial contamination. Traditional methods of the repair of mandible bone defects, such as the transfer of cells or tissue or bone grafts are not effective for the treatment of larger bone defects [41].

Autogenous bone grafts constitute a gold standard in surgery of the maxillofacial area. Excellent properties, such as osteoconductivity and osteoinductivity, of ceramic implants based on hydroxyapatite were confirmed also by our results. In the pig mandible, the ceramic plate promoted bone formation, remodelling and osteointegration, thus leading to enhanced repair of the cortical bone defect. Endosteal and periosteal regeneration was noticed, which was also confirmed by morphological and radiological evaluation and hardness examination. Our results also corroborated the claims of previous publications that described ceramic implants based on hydroxyapatite as a suitable material for the treatment of bone defects. The results of one study that compared two different types of materials for regeneration of the mandible, showed that 12 weeks after the application of a calcium phosphate biomaterial, it was almost completely degraded and the created defect was filled with a newly formed bone [42]. Our observations after 3 months from implantation of the HAP-based ceramic biomaterial to the site of the bone defect showed the presence of immature bone characterised by a bone matrix with a large number of osteoblasts, disarranged of collagen fibres and cement lines typical of the immature developing bone tissue. At this time, our macroscopical examination showed no residues of the ceramic plate at the defect site. Bioceramics, as a material with a brittle nature and high elastic modulus, do not seem to be the optimum choice for bone fixation [33,43]. The increase in bone density was noticed at the defect site after between 3 and 6 months of the healing period. Pearce et al. [44] stated that the bone composition, micro- and macrostructure, remodelling and density in the pig is very similar to that in humans. Wowern et al. [45] observed the relationship between the state of the periodontitis and the density of the bone in different body areas (mandible, forearm, femoral neck, lumbar vertebrae). They reported that the density of the mandible bone was significantly lower than that of bones in other investigated areas.

Our results show that the use of ceramic plates for the treatment of bone defects in mature pigs resulted in physiological values for the density of regenerated bones.

## 5. Conclusions

We can conclude that the success of marginal bone defect regeneration depends on many specific factors. There is the necessity of the presence of bone cells on the one hand and the use of a suitable biomaterial on the other hand. In our study, we implanted acellular biomaterial into the site of the created bone defects. The presence of blood in the surrounding bone tissue cells was very important for the proper formation of new bone; this relationship between the blood, bone tissue cells and the material (HAP plate) used in our study was very important. This fact should be emphasized, especially when solving problems with bone tissue and the therapy of bone defects in human medicine.

The future investigations dealing with bone defect treatments in the area of the maxillofacial surgery should focus on improving the biomaterials as scaffolds for mesenchymal stem cells. The implanted materials should be able to produce a suitable skeleton of newly formed regenerated bone and support its physiological potential and mechanical properties. Nowadays, many researchers involved in experimental medicine still use acellular scaffolds for the therapy of bone defects despite their demanding storage, low degree of survival of bone marrow cells and instability of growth factors or other components commonly used in tissue engineering.

## Figures and Tables

**Figure 1 biomedicines-11-00877-f001:**
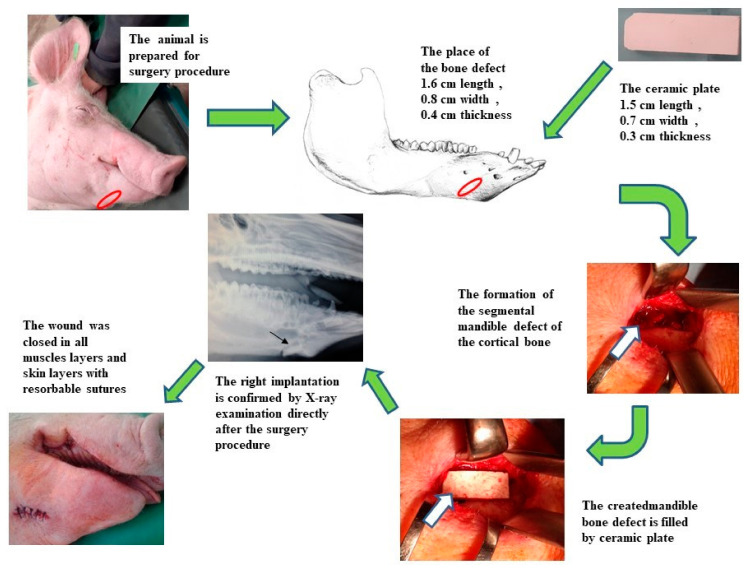
The scheme of the surgical procedure and postoperative evaluation of the HAP ceramic plate implantation.

**Figure 2 biomedicines-11-00877-f002:**
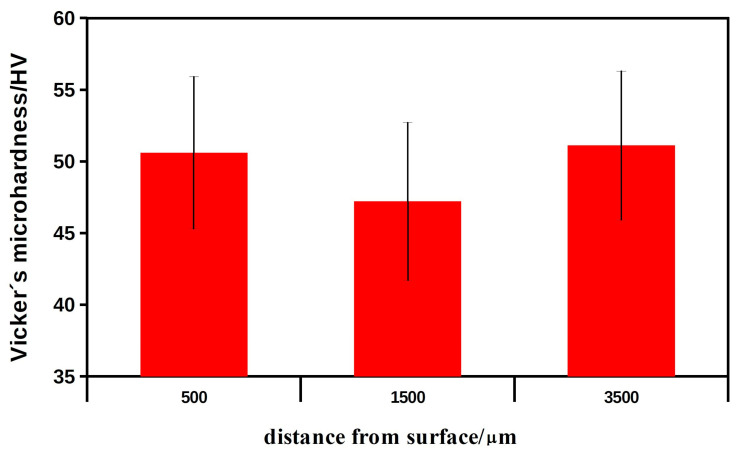
The values of Vicker microhardness of the new formed bone in the mandible at different distances from the bone surface with implanted HAP ceramic plates after 6 months of healing.

**Figure 3 biomedicines-11-00877-f003:**
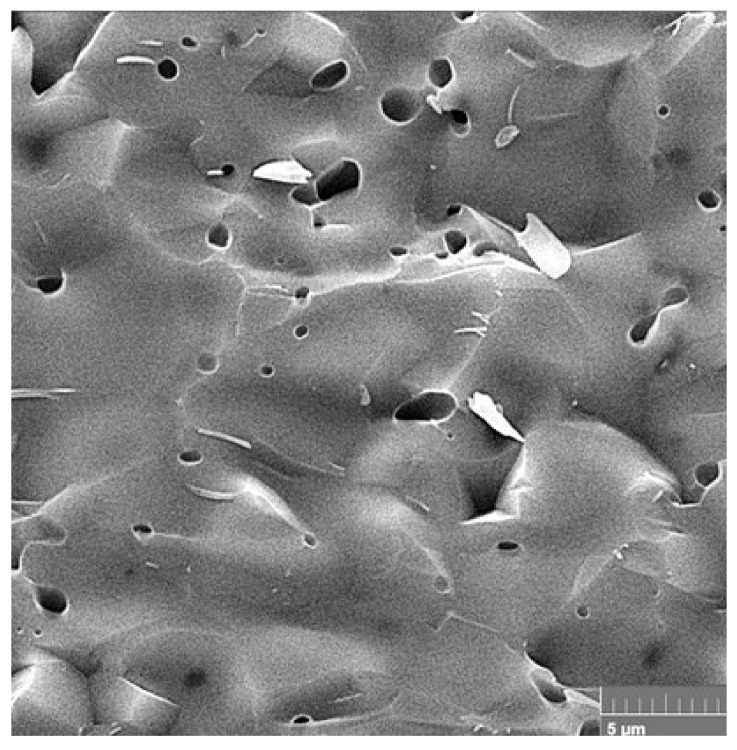
Microstructure of a fractured ceramic sample, where a low fraction of spherical pores is visible.

**Figure 4 biomedicines-11-00877-f004:**
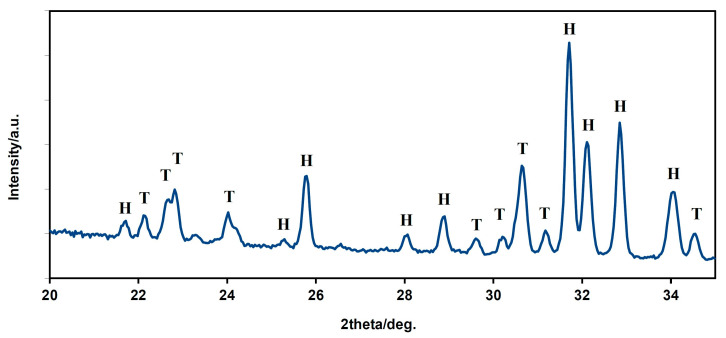
XRD patterns of HAP biphasic ceramics (H—hydroxyapatite, T—α TCP).

**Figure 5 biomedicines-11-00877-f005:**
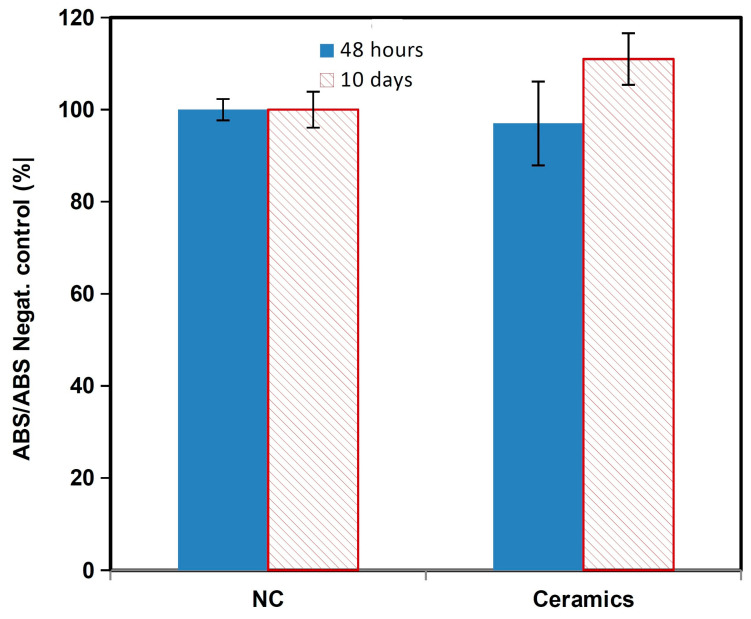
MTS cytotoxicity of the osteoblasts.

**Figure 6 biomedicines-11-00877-f006:**
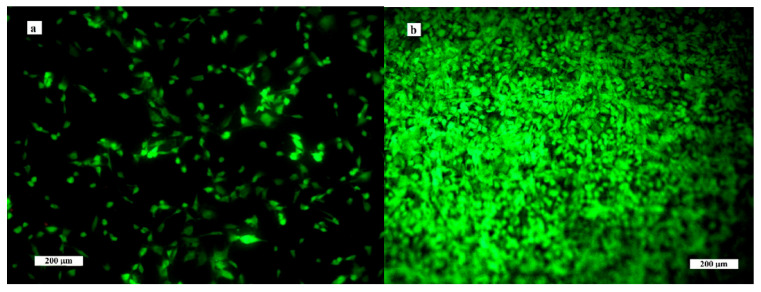
Distribution and morphology of osteoblasts cultured on ceramic samples for 2 (**a**) and 9 (**b**) days (live/dead staining).

**Figure 7 biomedicines-11-00877-f007:**
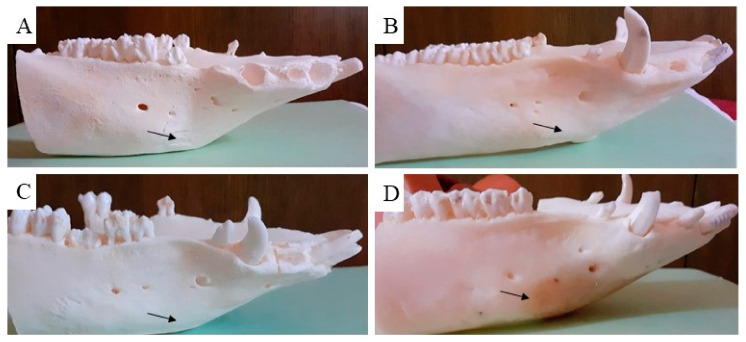
Macroscopic evaluation of the treated cortical bone defect after using the ceramic plate. The black arrow shows the place of the previous bone defect after treatment with the HAP ceramic plate in the time period (**A**) (3th), (**B**) (4th), (**C**) (5th) and (**D**) (6th month).

**Figure 8 biomedicines-11-00877-f008:**
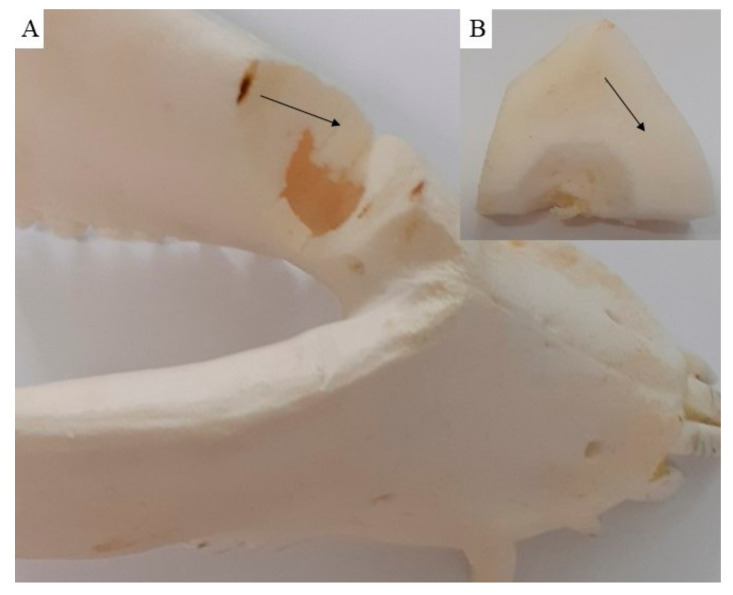
On the cross section (**A**), the new cortical bone (**B**) is visible, showing that the healing process resulted in uniform defect filling with new bone tissue (black arrow).

**Figure 9 biomedicines-11-00877-f009:**
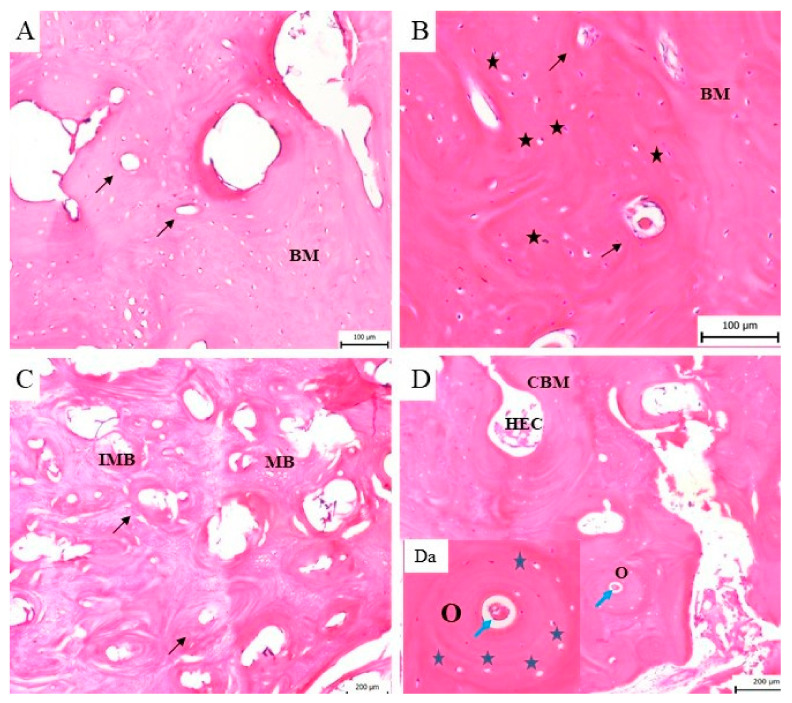
The histological evaluation of the bone formation. (**A**) Three months from implantation a new, immature bone with the bone matrix (BM) and typical cement lines (black arrow) is shown. (**B**) Four months after implantation the woven bone with a high number of osteoblasts (black asterisk) in the bone matrix (BM) and cement lines (black arrow) is still observable. (**C**) Five months after surgery, the bone matrix consists of area with immature, woven bone (homogeneous and fibrous structure) with cement lines (black arrow). In this sample differences between immature (IMB) and mature (MB) bone tissue are visible. (**D**) Six months after implantation the mature bone is fully developed and characterized by typical bone structures such as osteocytes, osteons (O), a cavity with the bone marrow (CBM) and haematopoietic cells (HEC) in the mineralised bone matrix. (**Da**) The osteon (O) is a structure of the mature bone, where a typical osteocyte (blue asterisk) arrangement with the Haversian system (blue arrow) in the middle is visible.

**Figure 10 biomedicines-11-00877-f010:**
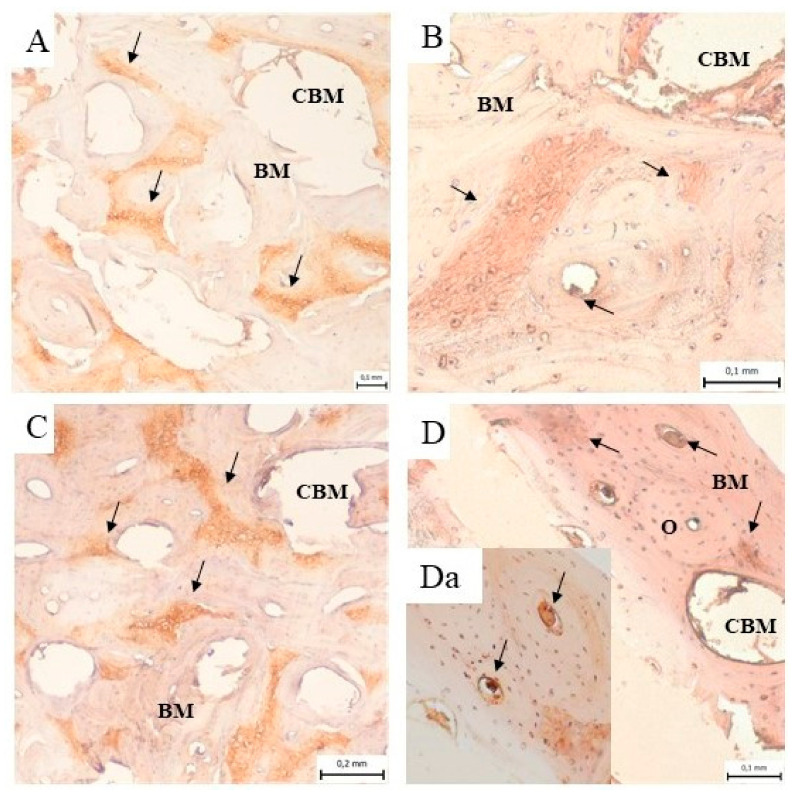
Immunohistochemical analysis of the samples from the cortical bone defect in a mandible treated with an HAP ceramic plate shows the presence of collagen I (black arrow) in the bone matrix (BM) of compact bone, in the spongy bone and around the bone marrow cavities (CMB) (**A**) 3 months, (**B**) 4 months, (**C**) 5 months and (**D**) 6 months after implantation. (**Da**) The presence of the expressed collagen I (black arrow) within and around the osteons (O).

**Figure 11 biomedicines-11-00877-f011:**
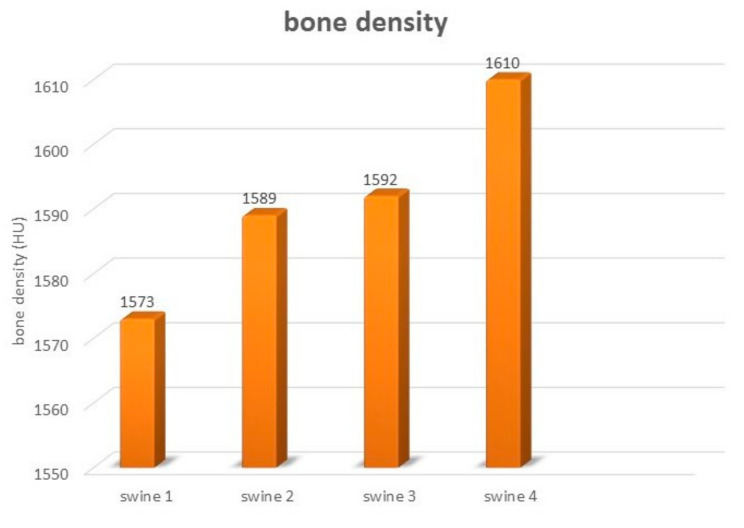
The bone density of the mandible in evaluated experimental animals.

**Figure 12 biomedicines-11-00877-f012:**
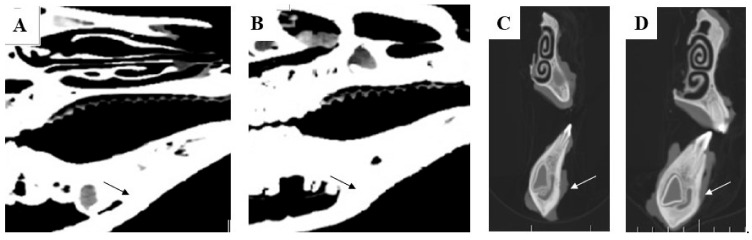
The computerized tomography section demonstrates high resolution of the sagittal plane (**A**,**B**), Pos: 0.3, Plane: (1.00, 0.00, −0.00, −0.28), W: 100, L: 35 and axial plane (**C**,**D**) Pos: −21.3, Plane: (0.00, −0.00, 1.00, 21.25), W: 100, L: 35, which proves the total regeneration of the cortical bone tissue (black and white arrows show the surgical area) after treating with the HAP ceramic plate.

**Figure 13 biomedicines-11-00877-f013:**
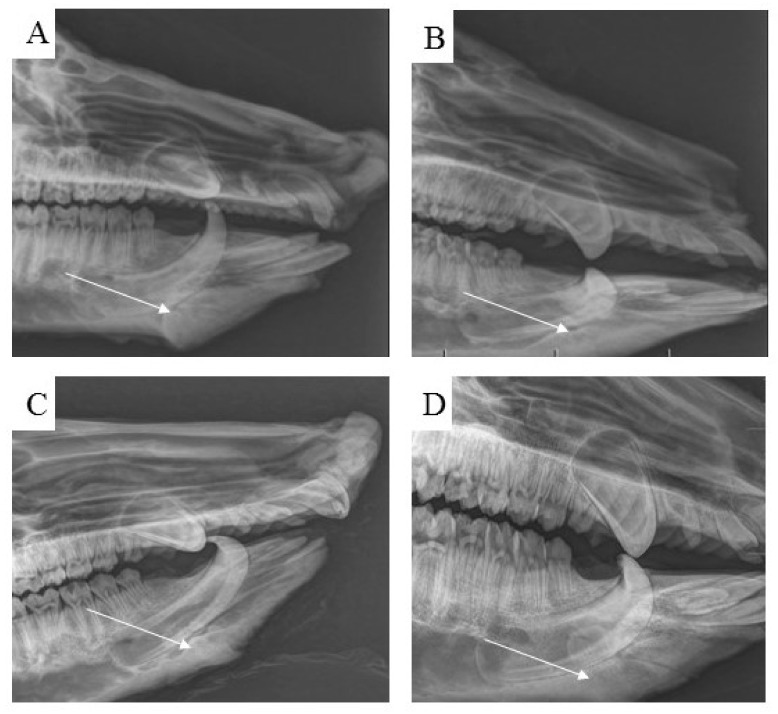
Using standard radiographs, we obtained the required spatial resolution of the right mandible at the level from the first to the second premolar of the evaluated area of the treated part of the cortical bone. For each time period, (**A**) 3 months, (**B**) 4 months, (**C**) 5 months and (**D**) 6 months, post implantation, the mandible defects were filled with newly formed bone tissue, which was completely integrated with the surrounding healthy bone.

**Table 1 biomedicines-11-00877-t001:** The properties of ceramic biomaterial (hydroxyapatite plate) used in our study.

Biocompatibility	Non-Toxic
Bioactivity	Biomaterial supported neo bone formation
Osteoinduction	Scaffold supported migration and proliferation of the mesenchymal stem cells
Osteoconduction	Biomaterial conducted the new bone formation
Biodegradation	Scaffold degradation
Bioresorption	Mandible defect was filled by the new regenerated bone
Mechanical resistance	Similar elastic and compressive strength to host bone
Porosity	Scaffold structure allowed neovascularization and growth of the stem cells

**Table 2 biomedicines-11-00877-t002:** The details of the animals used in our study.

Animals	Weight of the Animals (kg)	Material	Place of Defect	Size of Defect (cm)	Euthanasia of Animals after (Months)
Swine 1	258.9 kg	HAP ceramic plate	Cortical bone of the mandible body at the level of the P1–P2	1.6 × 0.8 × 0.4	3
Swine 2	262.4 kg	HAP ceramic plate	Cortical bone of the mandible body at the level of the P1–P2	1.6 × 0.8 × 0.4	4
Swine 3	257.2 kg	HAP ceramic plate	Cortical bone of the mandible body at the level of the P1–P2	1.6 × 0.8 × 0.4	5
Swine 4	263.8 kg	HAP ceramic plate	Cortical bone of the mandible body at the level of the P1–P2	1.6 × 0.8 × 0.4	6

## Data Availability

Data is contained within the article.

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
