# Peer review of "Regenerative Potential of Hydroxyapatite-Based Ceramic Biomaterial on Mandibular Cortical Bone: An *In Vivo* Study"

_biomedicines, 2023, doi:10.3390/biomedicines11030877_

Round 1
Reviewer 1 Report
The purpose of the current study was to evaluate the regeneration potential of a ceramic biomaterial based on hydroxyapatite obtained through an innovative method (as described by the authors). The introductory part is long and exhaustive where some aspects related to tested biomaterial are discussed which normally should be found in the results (for example table 1). The description of the materials should be done in chronological order. The part related to the in vitro characterization of the biomaterial is described in point 2.3 Evaluation of tissue regeneration (line 239) subsection 2.3.2 Characterization of HAP ceramics. In this subsection, normally other characteristics should be described (for example, the chemical composition, stability and other physical characteristics and not the degree of attachment and the cytotoxic potential of the biomaterial. Normally these aspects should be described within another sub point. Moreover, the assessment of the cytotoxic potential of biomaterials in contact with cells from the MC3T3E1 line is not fully described.
Why were females chosen for testing? what is the explanation for using adult animals? Why were bilateral defects not performed if the same animal was used as a control? Shouldn't the degree of regeneration of the defect be evaluated without the addition of a biomaterial? The post-interventional medication involved the administration of a non-steroidal anti-inflammatory drug. My question is if this medication does not influence the regeneration processes? Was the characterization of the biomaterial performed before or after the in vivo tests? as described in this article is not very clear. In the case of the cytotoxicity tests, why were the evaluations performed only after 48 h and respectively 9 days? After 48 hours was the acute toxicity evaluated and after 9 days the chronic one? How did you find out about cell viability? by using the LIVE/DEAD kit? Were cytomorphological changes observed in the cells attached to the cultivation surface?
Was the culture medium changed during this period? If you cultivated for 9 days, is there no possibility of contact inhibition? taking into account the fact that you used a substrate with aprox. 6mm in diameter and 0.5 mm in height. Would you also use controls for cytotoxicity tests?
Explanations are missing from the table, graph and figures except for figure 9 (line 472) where they appear. Figures 9 and 10 are unclear. Why is figure 8 followed by figure 10? (line 542).
Line 684 - the authors state that Resorption of the ceramic plate took about 1-3 months. Is this a general statement or does it strictly refer to this study?, taking into account that the first evaluation was carried out 3 months after the implantation of the biomaterial.
Line 726-728 Because the presence of blood containing cells of the surrounding bone tissue is very important for correct formation of new bone tissue, this relationship between blood with bone tissue cells and material (HAP plate) used in our study had to be considered also in our study - what does this statement refer to? The conclusions do not support the results.
Author Response
Response to Reviewer 1 Comments
Point 1: The purpose of the current study was to evaluate the regeneration potential of a ceramic biomaterial based on hydroxyapatite obtained through an innovative method (as described by the authors). The introductory part is long and exhaustive where some aspects related to tested biomaterial are discussed which normally should be found in the results (for example table 1). The description of the materials should be done in chronological order. The part related to the in vitro characterization of the biomaterial is described in point 2.3 Evaluation of tissue regeneration (line 239) subsection 2.3.2 Characterization of HAP ceramics. In this subsection, normally other characteristics should be described (for example, the chemical composition, stability and other physical characteristics and not the degree of attachment and the cytotoxic potential of the biomaterial. Normally these aspects should be described within another sub point. Moreover, the assessment of the cytotoxic potential of biomaterials in contact with cells from the MC3T3E1 line is not fully described.
Response 1: The introduction consists of many important information about topic. ,, aspects related to tested biomaterial are discussed which normally should be found in the results“ these characteristics and features of the biomaterial were described in previous publications, therefore are cited in our manuscript just. We wanted to acquaint readers about these aspects, because they are relevant for another practical using in our qualitative study, where was investigated the regenerative potential of this material on newly bone formation after creation of the marginal defect in the mandible. Some changes have been made to the introductory sections. We revised and changed section 2.3 and generally reorganized the entire manuscript. Additional data was added to the biomaterial section according to your recommendations. We also added the missing data on the cytotoxic potential of biomaterials in contact with cells from the MC3T3E1 line.
Point 2: Why were females chosen for testing?
Response 2: We selected only females because we did not want to mix the sexes. Gender is not a limiting criterion in the selection of our research. We do not think, that this criterion should affect the results of the study.
Point 3: What is the explanation for using adult animals?
Response 3: We used adult animals, because they have completed bone development, which does not affect the results of the formation and remodelling of the newly formed bone. Skeletal changes related to development and formation, as in young ones, cannot be expected.
Point 4: Why were bilateral defects not performed if the same animal was used as a control?
Shouldn't the degree of regeneration of the defect be evaluated without the addition of a biomaterial?
Response 4: The goal of our qualitative study was to evaluate the regenerative potential of ceramic biomaterial based on hydroxyapatite in formation and remodelling of the cortical bone in the lower jaw. First, we wanted to investigate the regenerative potential of the biomaterial compared to healthy bone tissue from the opposite side of the head. We are currently concluding our further results where we compared bilateral defects (one was filled with material, the other was left empty for spontaneous healing. In our research, we have several groups of animals that we want to compare. The next publication will be about the comparation and evaluation of bone defects filled with material, artificial cysts and healthy bone.
The post-interventional medication involved the administration of a non-steroidal anti-inflammatory drug. My question is if this medication does not influence the regeneration processes?
It is common knowledge that Flunixin is primarily used for the short-term treatment of moderate pain and inflammation. A study by Mohamed Elgendy et al. (Flunixin Meglumine Enhanced Bone Fracture Healing in Rabbits Associated with Activation of Early Collagen Deposition and Increased Expression of Vascular Endothelial Growth Factor 2021) aimed to evaluate the effects of flunixin meglumine on bone fracture healing in rabbits. A simple unilateral diaphyseal fracture was performed followed by K-wire fixation. Healing was assessed by radiography, histopathology and immunohistochemistry. Interestingly, the results revealed that FM improved bone fracture healing in combination with activation of early collagen deposition, marked angiogenesis, and increased vascular endothelial growth factor. In this case, it was a small laboratory animal rabbit, where a simple fracture occurred with only shifted fracture surfaces, which in our case is diametrically different from the marginal defect of the mandible in pigs, where is missing the part of the bone.
Point 5: Was the characterization of the biomaterial performed before or after the in vivo tests? as described in this article is not very clear.
Response 5: Characterization of the pure biomaterial was performed by in vitro tests before in vivo tests. The effect of the biomaterial in the mandibular bone was tested after in vivo studies when animal samples were available.
We have added the following additional information to the manuscript.
Point 6: In the case of the cytotoxicity tests, why were the evaluations performed only after 48 h and respectively 9 days? After 48 hours was the acute toxicity evaluated and after 9 days the chronic one? How did you find out about cell viability? by using the LIVE/DEAD kit? Were cytomorphological changes observed in the cells attached to the cultivation surface?
Response 6: The hydroxyapatite ceramics is uded for long time as bone filler and it is extremely known fact that in such a form – microcrystalline hydroxyapatite – the contact cytotoxicity is very seldom due to relative smooth surface topography as well as sufficiently large area of grains where cell can be good adhered. Note that it is key fact which is in opposite to nanocrystalline hydroxyapatite with needle-like morphology which is frequently found e.g. in cements. Moreover, we added to text standard cytotoxicity evaluation according to ISO 10993-5 (2003) which clearly identified non cytotoxicity of samples. From the point of view selected testing time – it is clear that in vitro testing on samples has only significance as far as the cell grows in monolayer but as can be seen in live/dead stained images, thick cell multilayer was observed after 9 days and further prolongation of cultivation time is unnecessary.
Point 7: Was the culture medium changed during this period? If you cultivated for 9 days, is there no possibility of contact inhibition? taking into account the fact that you used a substrate with aprox. 6mm in diameter and 0.5 mm in height. Would you also use controls for cytotoxicity tests?
Response 7: In standard ISO 10993-5 (2003) is clearly described how controls can be used for evaluation of in vitro cytotoxicity – in our case, the wells in microplate (free of sample) were used as control.
Point 8: Explanations are missing from the table, graph and figures except for figure 9 (line 472) where they appear. Figures 9 and 10 are unclear. Why is figure 8 followed by figure 10? (line 542).
Response 8: In general, we have made the necessary changes to the figures, graphs and tables as per your recommendations. We adjusted the order of the images (for example figure 8 and 10). We revised and changed individual chapters and overall reorganized the entire manuscript.
Point 9: Line 684 - the authors state that Resorption of the ceramic plate took about 1-3 months. Is this a general statement or does it strictly refer to this study?, taking into account that the first evaluation was carried out 3 months after the implantation of the biomaterial.
Response 9: It is a statement arising from the results of our qualitative study. Now, that we know that HAP ceramic has regenerative potential in newly bone tissue formation and its complete resorption was notice after 3 months after implantation, we will continue our research and focus on monitoring this potential in a shorter time interval than 3 months after implantation. Our results show, that macroscopically new cortical bone was fully formed after 3 months, although at the microscopic level we still observed immature bone that continued to develop and remodel during this period. Mature bone was observed up to 6 months after implantation. All investigative methods used in our study confirm complete resorption of the HAP ceramic plate after 3 months.
Point 10: Line 726-728 Because the presence of blood containing cells of the surrounding bone tissue is very important for correct formation of new bone tissue, this relationship between blood with bone tissue cells and material (HAP plate) used in our study had to be considered also in our study - what does this statement refer to? The conclusions do not support the results.
Response 10: This statement refers to the fact that in our quantitative study we put a huge emphasis on fulfilling this fact, aspect as much as possible. Because, just as it has an enormous impact on bone healing and regeneration in in vivo studies in large animal models, this fact should be emphasized especially when solving problems with bone tissue and therapy of bone defects in human medicine. The conclusion has been modified according to your recommendations. Several studies point to the importance of the presence of blood in the healing process of bone tissue:
|
1. Massimo Marenzana, Timothy R. Arnett: The Key Role of the Blood Supply to Bone, Bone Research (2013) 3: 203-215. 2. Joanna Filipowska, Krzysztof A. Tomaszewski, Łukasz Niedzwiedzki, Jerzy A. Walocha, Tadeusz, Niedzwiedzki: The role of vasculature in bone development, regeneration and proper systemic functioning, Angiogenesis (2017) 20:291–302.. 3. Guanyin Zhu, Tianxu Zhang, Miao Chen, Ke Yao, Xinqi Huang, Bo Zhang, Yazhen Li, Jun Liu, Yunbing Wang, Zhihe Zhao: Bone physiological microenvironment and healing mechanism: Basis for future bone-tissue engineering scaffolds, Bioactive Materials 6 (2021) 4110–4140
|
Thank you for rating our study. We greatly appreciate the overall rating as well as all comments on the manuscript. We have included all your comments in editing our manuscript. According to your recommendation, we have edited the entire manuscript. We have added information on biomaterials, divided into in vitro and in vivo studies. We redesigned the images, added captions.

Reviewer 2 Report
Authors present a research article focused on the fabrication and characterization hydroxyapatite-based ceramic biomaterial. Then they study regenerative potential of such biomaterial through in vitro cell culture with osteoblast and on mandibular cortical bone in-vivo using pig as animal model. The present work would be useful for researchers working on production new biomaterials based on HAP for biomedical applications. My comments are below.
My remarks:
1.- I would suggest to remove from the title “innovative” and to include italic for “in vivo”. Indeed, the HAP are the older ceramics studied so far as biomaterial.
2.- The article must be written in third person. This must be corrected through the whole manuscript. Article has spaces, comma and few typo mistakes.
3.- figure 1 is blurry, unreadable.
4.- figure 3 labels are unclear.
5.- Not replicates were performed during the in vivo studies?
6.- why authors decide to use pig, they already use this material through in vitro cell cultures? And with other small animals?
7.- figure 2 is not clear. We have 3 bars corresponding to three measurements “data were collected after the respective observation periods (3th, 4th, 5th and 6th month)”???
8.- Authors mix phisco-chemical characterization with biological studies. I will recommend to split the biological from the non-biological assays. For instance, they shown osteoblast cell culture in figure 5, but not of this was separated in materials and methods.
9.- I would suggest to remove the details of the SEM from figure 3. Besides, authors are confident about the use of 2019 picture for this work in 2023?
10. All the histological images are hard to follow. Suggest to include arrow in the ROI, remove the scale from the picture which is unreadable.
11.- figure 8 blurry. Overall, very bad images, blurry, no scales, unreadable.
12.- Not relevant and recent bibliography is reported. Only 2 paper from 2022 the rest of the cited work are older.
In general, the work presented include nice experimental data and analysis. But overall, is not well structured, and need to be fully revised if editor consider to publish this paper. Not only the figure, but the outline, the organization, the English and typo mistakes.
Author Response
Response to Reviewer 2 Comments
Point 1: I would suggest to remove from the title “innovative” and to include italic for “in vivo”. Indeed, the HAP are the older ceramics studied so far as biomaterial.
Response 1: The title of the manuscript has been modified according to your suggestions.
Point 2: The article must be written in third person. This must be corrected through the whole manuscript. Article has spaces, comma and few typo mistakes.
Response 2: A native speaker (who is a co-author in our article) checked the entire manuscript once more and the manuscript has been edited according to your suggestions.
Point 3: figure 1 is blurry, unreadable.
Response 3: We have modified Figure 1 based on your recommendations. The image should now be readable and not blurred.
Point 4: figure 3 labels are unclear.
Response 4: The label to the figure 3 was edited. All important information to this figure and method are present in the text, in the point 3.2 Microstructure, compressive strength and XRD phase analysis of ceramics.
Point 5: Not replicates were performed during the in vivo studies?
Response 5: The goal of our qualitative study was to evaluate the regenerative potential of ceramic biomaterial based on hydroxyapatite in the formation and remodeling of cortical bone in the lower jaw. In this part of the research, we focused on the qualitative potential of the material for new bone formation, its resorption time and its effect on the formation and remodeling of cortical bone. On the basis of these results, we can continue further and focus our research on quantitative analysis in the monitoring of individual groups of experimental animals.
Point 6: why authors decide to use pig, they already use this material through in vitro cell cultures? And with other small animals?
Response 6: Compared to other farm animals, the domestic pig is considered an ideal animal model that is applicable in a wide range of biomedical research. This species of animal reaches sexual maturity in a relatively short time, has a short gestation period and a considerable litter size. Pigs are bred for a longer period of time, which makes them a suitable animal model for biomedical research and for longer-term studies. Thanks to these advantages as well as similar anatomical or physiological properties and also their size (pigs have a similar size and body weight to an adult human), this species is often used as a model animal for organ transplants (heart, lungs, kidneys, skin) as well as other experimental , surgical interventions, preclinical tests of various pharmacological preparations or numerous naturally occurring diseases in the human population.
The swine may serve as a very good animal model for the testing of bone implants, because the bone mineral density, morphology, remodelling rate and healing are similar to that in humans. The immature and mature swine skeletons have individual growing phases with very well developed Haversian systems. Porcine models were used in the studies of the osteogenic potential of the biomaterials in maxillofacial area of the head and in dental surgery involving teeth implants.
In vitro testing, as well as in vivo testing on small laboratory animals (mouse, rat, rabbit), or other alternative methods, such as testing material properties on a chicken embryo, will not provide us with sufficient and relevant results and a real picture of the monitored issue.
Point 7: figure 2 is not clear. We have 3 bars corresponding to three measurements “data were collected after the respective observation periods (3th, 4th, 5th and 6th month)”???
Response 7: The figure 2 and the accompanying information have been modified and supplemented.
Point 8: Authors mix phisco-chemical characterization with biological studies. I will recommend to split the biological from the non-biological assays. For instance, they shown osteoblast cell culture in figure 5, but not of this was separated in materials and methods.
Response 8: There have been several changes in parts of the material, methods and results. We revised and changed the section where biomaterial characteristics and properties were described. Biological and non-biological tests were divided. Additional data has been added to the biomaterials section as per your recommendations. We also added the missing data on the cytotoxic potential of biomaterials in contact with MC3T3E1 cells related to Figure 5.All changes are included in the text of the manuscript.
Point 9: I would suggest to remove the details of the SEM from figure 3. Besides, authors are confident about the use of 2019 picture for this work in 2023?
Response 9: Based on your recommendations, we have made changes to Figure 3. The authors are confident about the use of 2019 image is appropriately used for this work in 2023. The entire research has been ongoing for years. The entire in vivo study was preceded by years of in vitro study, followed by further steps summarizing and evaluating the results. Therefore, this image is relevant and suitable for this manuscript as well. Of course, we will continue our research. We have further goals and improvements in the process of solving the issue of using biomaterials in the therapy of hard tissues (bones, cartilage).
Point 10: All the histological images are hard to follow. Suggest to include arrow in the ROI, remove the scale from the picture which is unreadable.
Response 10: We redid the histological images and adjusted them according to your recommendations. We added the required characteristics to the images for better orientation to make them easier to read.
Point 11: figure 8 blurry. Overall, very bad images, blurry, no scales, unreadable.
Response 11: All images have been edited, supplemented for better readability as per your suggestion.
Point 12: Not relevant and recent bibliography is reported. Only 2 paper from 2022 the rest of the cited work are older.
Response 12: The list of our bibliography corresponds to the studied issue. We used such sources that we could compare with our methods, procedures and also the type of acellular biomaterial used in our study. If you would like to recommend other relevant sources to cite in our manuscript, we have no problem listing them there.
Point 13: In general, the work presented include nice experimental data and analysis. But overall, is not well structured, and need to be fully revised if editor consider to publish this paper. Not only the figure, but the outline, the organization, the English and typo mistakes.
Response 13: Thank you for rating our study. We greatly appreciate the overall rating as well as all comments on the manuscript. We have included all your comments in editing our manuscript. According to your recommendation, we have edited the entire manuscript. We have added information on biomaterials, divided into in vitro and in vivo studies. We redesigned the images, added captions. Regarding English, the entire manuscript was proofread in English before first submission. Our co-author, who is a native speaker, has also repeatedly proofread the English language once more now.

Round 2
Reviewer 1 Report
I believe that the authors substantially corrected the initial manuscript and took into account my recommendations. In this form it can be accepted for publication.
Reviewer 2 Report
author's responded reveiwer comments